# How Rhizosphere Microbial Assemblage Is Influenced by Dragon Fruits with White and Red Flesh

**DOI:** 10.3390/plants13101346

**Published:** 2024-05-13

**Authors:** Xinyan Zhou, Siyu Chen, Lulu Qiu, Liyuan Liao, Guifeng Lu, Shangdong Yang

**Affiliations:** 1Guangxi Key Laboratory of Agro-Environment and Agro-Products Safety, National Demonstration Center for Experimental Plant Science Education, Agricultural College, Guangxi University, Nanning 530004, China; zhouxinyan0923@163.com (X.Z.); chensiyu233@mails.ucas.ac.cn (S.C.); 18778409037@163.com (L.Q.); 15278815458@163.com (L.L.); 2Horticulture Research Institute, Guangxi Academy of Agricultural Sciences, Nanning 530007, China

**Keywords:** dragon fruit (*Hylocereus undulatus* Britt), flesh color, microbes, high-throughput sequencing

## Abstract

The synthesis of betalain using microorganisms is an innovative developmental technology, and the excavation of microorganisms closely related to betalain can provide certain theoretical and technical support to this technology. In this study, the characteristics of soil microbial community structures and their functions in the rhizospheres of white-fleshed dragon fruit (*Hylocereus undatus*) and red-fleshed dragon fruit (*Hylocereus polyrhizus*) were analyzed. The results show that the soil bacterial and fungal compositions in the rhizospheres were shaped differently between *H. undatus* and *H. polyrhizus*. Bacterial genera such as *Kribbella* and *TM7a* were the unique dominant soil bacterial genera in the rhizospheres of *H. undatus*, whereas *Bradyrhizobium* was the unique dominant soil bacterial genus in the rhizospheres of *H. polyrhizus*. Additionally, *Myrothecium* was the unique dominant soil fungal genus in the rhizospheres of *H. polyrhizus*, whereas *Apiotrichum* and *Arachniotus* were the unique dominant soil fungal genera in the rhizospheres of *H. undatus*. Moreover, *TM7a*, *Novibacillus*, *Cupriavidus*, *Mesorhizobium*, *Trechispora*, *Madurella*, *Cercophora*, and *Polyschema* were significantly enriched in the rhizospheres of *H. undatus*, whereas *Penicillium*, *Blastobotrys*, *Phialemonium*, *Marasmius*, and *Pseudogymnoascus* were significantly enriched in the rhizospheres of *H. polyrhizus*. Furthermore, the relative abundances of Ascomycota and *Penicillium* were significantly higher in the rhizospheres of *H. polyrhizus* than in those of *H. undatus*.

## 1. Introduction

Dragon fruit can be categorized into two varieties based on the color of its flesh, distinguished as white (*Hylocereus undatus*) and red (*Hylocereus polyrhizus*) flesh [1,2]. *H. polyrhizus* is rich in betalain, which is a water-soluble natural plant pigment [3]. In contrast, *H. undatus* lacks betalain. Therefore, *H. polyrhizus* has an advantage over *H. undatus*, contributing to minerals, proteins, phenolics, and antioxidant properties [4,5,6,7]. Due to the gradual appreciation of betalain as a food additive and its potential medicinal value, based on biotechnologies, producing betalain has become a hot spot for researchers.

Rhizospheric soil microbial community composition is closely related to the plant variety [8,9], and different plant varieties produce diverse compositions by recruiting various soil microbial communities in the rhizospheres [10,11]. It has been demonstrated that the soil microbial community structure significantly differs in rhizospheres among poplar varieties [12]. Additionally, soil microbial community structures not only significantly differ in the rhizospheres of different soybean varieties but, also, each soybean variety has specific soil microbial compositions and networks in different rhizospheres [13,14]. Moreover, soil microbiomes in the rhizospheres of olives [15] and pumpkins [16,17] differ between different varieties. 

Previous studies have confirmed that the rhizospheric soil microbial community interacts with plant varieties [18,19]. Firstly, the functional requirements of plants play a crucial role in the assembly of rhizospheric microbial communities [20]. Plant varieties can not only shape the soil microbial community structures in rhizospheres but also affect the soil microbial functions in rhizospheres [21]. For instance, studies have reported that soil microbial compositions and functions in the rhizospheres of strawberries significantly depend on the variety, and disease-resistant and high-yielding strawberry varieties commonly recruit more beneficial soil microbes [22]. Additionally, a mulberry variety resistant to bacterial wilt was reported to be enriched with fungal communities that can prevent the spread of soil-borne diseases, enhance plant drought resistance, and provide tolerance against biological or abiotic stresses [23]. Chen et al. [24] found that different rhizospheric soil microbes are recruited by tomatoes with different fruit color phenotypes. Xiao et al. [25] also found differences in rhizospheric microbes recruited by different watermelon phenotypes relating to rind color formation. In addition, plant traits were also significantly influenced by soil microorganisms [26], such as disease resistance [27], root architecture [28], and drought tolerance [29], all of which are modified by soil microbes in rhizospheres.

Additionally, soil microorganisms are also able to produce various phytohormones related to fruit pigment production and metabolite accumulation, some of which are also closely related to the synthesis of betalain [30]; for example, cyanobacteria serve as the favored origin of exogenous cytokinin [31]. These cytokinins play a crucial role in enhancing the accumulation of betalain by activating the dopa oxidase protein and binding it to tyrosine [32]. *Botrytis* fungi can produce abscisic acid [33], *Aspergillus* fungi can synthesize gibberellins [34], and abscisic acid and gibberellins inhibit betalain biosynthesis [35]. *B. rhodina* has the potential to produce jasmonate under controlled conditions [36]. Jasmonate can be induced by ultraviolet-B irradiation and can promote betalain accumulation [37]. Additionally, the biosynthesis pathways of betalain were found to be associated with *Streptomyces*, *Bacillus megaterium*, *Agaricus bisporus*, and *Hygrocybe* [38,39,40,41]. Research on the synthesis of betalain using microorganisms has also been developed. For example, Grewal et al. [42] achieved the microbial synthesis of betalain from glucose using *Saccharomyces cerevisiae*; Wang et al. [43] used biofilm surface fermentation to produce betalain from *Penicillium novae-zelandiae*. The fruits of red-fleshed dragon fruit (*H. polyrhizus*) contain abundant betalain, but whether soil microbes in rhizospheres contribute to betalain synthesis is unknown, as, to date, there are no reports on this. Therefore, the differences in soil microbial compositions in the rhizospheres of white-fleshed dragon fruit (*H. undatus*) and red-fleshed dragon fruit (*H. polyrhizus*) were analyzed.

## 2. Materials and Methods

### 2.1. Experimental Site

This experiment was conducted at the vegetable base of the Agricultural College, Guangxi University, Nanning, Guangxi, P.R. China (108°17′25″ E, 22°51′02″ N). Twenty-five plants each of the varieties ‘Nanning white-flesh’ (*H. undatus*, RW) and ‘YuanzhiHong red-flesh’ (*H. polyrhizus*, RR) (Figure 1) were planted in pots (40 cm high and 35 cm in diameter) in October 2021, one plant in each pot.

The soil physicochemical properties were as follows: soil pH 5.84; organic matter content 9.46 g·kg^−1^; total nitrogen, phosphorus, and potassium 0.62 g·kg^−1^, 0.76 g·kg^−1^, and 7.83 g·kg^−1^, respectively. The available nitrogen, phosphorus, and potassium contents were 16.36 mg·kg^−1^, 0.56 mg·kg^−1^, and 86.35 mg·kg^−1^, respectively.

### 2.2. Soil Sample Collection

The samples were collected in July 2022. Rhizospheric soil samples were collected using the shaking-off method. Briefly, three dragon fruit plants of each variety were randomly selected and loosened to approximately 25 cm in diameter using a sterile shovel, after which the base of each plant was pulled up entirely with the soil. The soil adhering to the dragon fruit roots was carefully collected as the rhizospheric soil in a sterile bag. Soil samples were also collected from the same field without any plant growth treatment as background soil (CK) and then transported to the laboratory as quickly as possible. After the samples were passed through a 2 mm stainless-steel sieve, the sifted soil was stored at −80 °C.

### 2.3. Test Methods

Total DNA extraction and polymerase chain reaction (PCR) amplification of the plant rhizosphere samples were conducted as follows: Total DNA extraction was performed according to the instructions of the Fast DNA^®^ Spin Kit for Soil (MP Biomedicals, Thomas Irvine, CA, USA), and the DNA concentration and purity were determined using a NanoDrop 2000 spectrophotometer (Thermo Fisher Scientific, Waltham, MA, USA). The extracted DNA was examined on a 1% agarose gel, the quality of the extracted DNA was detected via gel electrophoresis, and PCR amplification was performed on an GeneAmp^®^ 9700 (ABI, Los Angeles, CA, USA) using the extracted rhizosphere microbial DNA as a template.

Illumina MiSeq sequencing was performed as follows: PCR products from the same sample were recovered using a gel recovery kit, purified by elution with Tris-HCl buffer, and then recovered using a 2% agarose gel for detection and quantification of the recovered products using a Quantus™ Fluorometer (Promega, Madison, MI, USA). Illumina library construction was performed using purified amplified fragments. Sequencing was performed using Illumina’s MiSeq PE platform. The bacterial 16S rRNA gene was amplified using the primer pair 338F (5′-ACTCCTACGGGAGGCAGCAG-3′) and 806R (5′-GGACTACHVGGGTWTCTAAT-3′). For the amplification of fungal ITS regions, the primer pair ITS1F (5′-CTTGGTCATTTAGAGGAAGTAA-3′) and ITS2R (5′-GCTGCGTTCTTCATCGATGC-3′) was employed. The raw data for soil bacterial and fungal sequences were deposited in the NCBI Sequence Read Archive (SRA) database under accession numbers PRJNA1066732 and PRJNA1066751, respectively.

The paired-end (PE) reads obtained from Illumina sequencing were first spliced according to the overlap relationship, while the sequences were quality-controlled and filtered, and the samples were differentiated and then analyzed through clustering and species taxonomy using the I-sanger cloud data analysis platform (http://www.majorbio.com) of the Majorbio Bio-Pharm Technology Co., Ltd. (Shanghai, China). Based on the taxonomic information, statistical analysis of the community structure was performed at each taxonomic level. The operational taxonomic unit (OTU) clustering steps were as follows: Utilizing Uparse software (version 7.0.1), non-redundant sequences were extracted from optimized sequences to facilitate a reduction in the redundant computational burden in the intermediate analysis processes. The non-redundant sequences (excluding single sequences) were clustered into OTUs based on 97% similarity, and chimeras were removed during the clustering process to obtain representative sequences of the OTUs. Subsequently, all optimized sequences were mapped to the representative sequences of the OTUs, and sequences with a similarity of 97% or higher to the OTU representative sequences were selected to generate the OTU table. Taxonomic analysis of representative sequences of OTUs at a 97% similarity level was conducted using the RDP Classifier version 2.11 on the QIIME platform (http://qiime.org/scripts/assign_taxonomy.html, accessed on 17 March 2023). A confidence threshold of 0.7 was applied to obtain species classification information corresponding to each OTU. Bacteria and fungi were individually aligned against the Silva database (Release 138, http://www.arb-silva.de) and the Unite database (Release 8.0, http://unite.ut.ee/index.php, accessed on 17 March 2023) for comparative analysis.

### 2.4. Statistical Analysis

The experimental data underwent statistical analysis using Excel 2019 and SPSS Statistics 18 software. Data are presented as the mean ± standard deviation (SD). The analysis of the rhizosphere microbial community structure encompassed alpha diversity analysis of bacteria and fungi, so we utilized the Shannon and Invsimpson indices to estimate the microbial diversity, and the Ace and Chao indices were used to represent species richness. Additionally, we conducted taxonomic analyses at the phylum and genus levels for rhizosphere microbial communities, exploring distinctions between the rhizosphere microbial community structures of *H. undatus* and *H. polyrhizus*.

## 3. Results

### 3.1. Soil Microbial Diversity in the Rhizospheres of Dragon Fruits with White and Red Flesh

As shown in Table 1, the coverage indices reached at least 98%, indicating that the results may effectively reflect the true situation of the species. The soil bacterial diversity indices, i.e., the Shannon and Invsimpson indices, were not significantly different in the rhizospheres of the two dragon fruits with white and red flesh and neither were the soil bacterial richness indices, i.e., the Ace and Chao indices. Moreover, the soil fungal diversity and richness in the rhizospheres between the two dragon fruits with white and red flesh also displayed the same trends as those of the soil bacterial community.

Principal coordinates analysis (PCoA) based on the OTU level (Bray–Curtis distance, Adonis test) revealed that, for soil bacteria in RW, RR, and CK, the Adonis test results were R^2^ = 0.3785, *p* = 0.005 (Figure 2A); for fungi, the Adonis test results were R^2^ = 0.6406, *p* = 0.005 (Figure 2C). All of the above results indicate that the differences between treatment groups were greater than those within treatment groups. Additionally, these findings suggest that the data analysis was reliable. Meanwhile, it also indicates that there were highly significant differences (*p* < 0.01) in the community composition of rhizosphere bacteria and fungi among RW, RR, and CK.

Moreover, partial least squares discriminant analysis (PLS-DA) also showed that the soil bacterial (Figure 2B) and fungal (Figure 2D) compositions at the OTU level clustered separately in RW, RR, and CK. Furthermore, based on the dispersion of the sample point distributions, each group of samples clustered together with relatively little variation in community structure within the group.

As an example, the soil microbial communities in rhizospheres of RW and RR exhibited significant differences.

### 3.2. Soil Microbial Community Structure in the Rhizospheres of Dragon Fruits with White and Red Flesh

As shown in Figure 3A, the common dominant endophytic bacterial phyla (the relative abundances were higher than 1%) in RW and RR were Actinobacteriota (28.95–30.83%), Pseudomonadota (25.05–26.01%), Firmicutes (12.17–9.98%), Chloroflexi (10.27–11.25%), Acidobacteriota (6.27–7.16%), Gemmatimonadota (4.56–5.02%), Bacteroidota (3.91–3.43%), Patescibacteria (2.80–1.01%), and Myxococcota (2.03–1.74%) (The first value corresponds to RW and the second to RR, the same below).

The common dominant endophytic bacterial genera (the relative abundances were higher than 1%) in RW and RR were *Actinomadura* (3.92–4.03%), *Sphingomonas* (3.64–3.95%), *Saccharomonospora* (3.31–3.25%), *Bacillus* (2.09–1.71%), *Novibacillus* (1.78–1.37%), *Gemmatimonas* (1.58–1.58%), *Streptomyces* (1.38–1.21%), *Haloactinopolyspora* (1.19–1.21%), *Knoellia* (1.16–1.16%), and *Acidibacter* (1.01–1.02%) (Figure 3B).

Further, *Kribbella* (1.33%) and *TM7a* (1.11%) were the unique dominant soil bacterial genera in the rhizosphere of RW, whereas *Bradyrhizobium* (1.05%) was the unique dominant soil bacterial genus in the rhizosphere of RR. 

In addition, the common dominant endophytic fungal phyla in RW and RR were Ascomycota (51.28–73.75%), Basidiomycota (40.33–18.10%), Mortierellomycota (2.62–3.06%), and Chytridiomycota (2.22–1.44%) (Figure 4A). 

At the genus level, the common dominant endophytic fungal genera in RW and RR were *Trechispora* (24.93–9.54%), *Trichocladium* (13.30–8.35%), *Saitozyma* (11.55–4.27%), *Penicillium* (9.72–29.63%), *Talaromyces* (4.21–6.00%), *Mortierella* (2.57–2.97%), *Trichoderma* (2.33–2.77%), *Aspergillus* (2.26–4.22%), *Chaetomium* (1.56–2.50%), and *Apiotrichum* (1.10–1.49%) (Figure 4B). 

Additionally, *Arachniotus* (1.02%) was the unique dominant soil fungal genus in the rhizosphere of RW, whereas *Myrothecium* (2.58%) was the unique dominant soil fungal genus in the rhizosphere of RR. 

Linear discriminant analysis effect size (LEfSe) analysis with a linear discriminant analysis (LDA) threshold of 3.0 was employed to investigate the significant variances and primary influential biomarker categories between bacterial communities inhabiting the rhizosphere soils of RW and RR. A higher LDA score indicates a greater impact on species abundance under differential effects. 

As shown in Figure 5, *TM7a*, *Novibacillus*, *Cupriavidus,* and *Mesorhizobium* were significantly enriched in the rhizospheres of RW.

Moreover, *Trechispora*, *Madurella*, *Cercophora*, and *Polyschema* were significantly enriched in the rhizospheres of RW, whereas *Penicillium*, *Blastobotrys*, *Phialemonium*, *Marasmius*, and *Pseudogymnoascus* were significantly enriched in the rhizospheres of RR (Figure 6).

### 3.3. Functional Prediction of Soil Microorganisms in the Rhizospheres between the Two Dragon Fruits with White and Red Flesh

Phylogenetic investigation of communities by reconstruction of unobserved states 2 (PICRUSt2) functional prediction was used to analyze the expression of the soil bacteria between RW and RR (Figure 7). The top 10 pathways were aerobic respiration Ⅰ (cytochrome c), pyruvate fermentation to isobutanol (engineered), L-isoleucine biosynthesis Ⅱ, L-isoleucine biosynthesis Ⅰ (from threonine), L-valine biosynthesis, super pathway of branched amino acid biosynthesis, fatty acid salvage, TCA cycle V (2-oxoglutarate/ferredoxin oxidoreductase), TCA cycle Ⅰ (prokaryotic), and L- isoleucine biosynthesis Ⅲ. Additionally, a Wilcoxon rank sum test was used to assess the soil bacterial expression between the rhizospheres of RW and RR. The results show that there was no significant difference in the expression of soil bacteria in each pathway between the rhizospheres of the dragon fruits with white and red flesh. However, the expression of bacteria in the rhizosphere of the 10 RR-related MetaCyc pathways was higher than that of RW.

FUNGuild software (http://www.funguild.org/, accessed on 20 March 2023) was also used to predict the functions of the soil fungal communities in the rhizospheres of RW and RR. The results show that soil fungi could be divided into three types based on their nutrition modes, i.e., Symbiotroph, Pathotroph, and Saprotroph. Furthermore, based on the fungal community pathways, fungi in the rhizospheres of RW and RR can be classified into nine guilds, i.e., Undefined Saprotroph, unknown; Fungal Parasite-Undefined Saprotroph; Animal Pathogen-Dung Saprotroph-Endophyte-Epiphyte-Plant Saprotroph-Wood Saprotroph, Wood Saprotroph, Endophyte-Litter Saprotroph-Soil Saprotroph-Undefined Saprotroph, Plant Pathogen, Animal Pathogen, Soil Saprotroph, and others. In comparison with those of soil fungi in the rhizospheres of RW, the relative abundances of soil fungi in the rhizospheres of RR associated with the Fungal Parasite-Undefined Saprotroph and Wood Saprotroph guilds decreased; however, the relative abundances of soil fungi associated with the Saprotroph guild in RR were greater than in those in RW (Figure 8).

## 4. Discussion

Different plant varieties have a direct impact on the soil microbial community in rhizospheres [44]. This is attributed to the fact that diverse plant roots secrete distinct substances and specific molecular signals [45,46,47]. Consequently, the compositions of soil microbes can be influenced by different plants [48], leading to the enrichment of diverse soil microbes in specific microenvironments, such as the rhizospheres of various plants [49,50,51]. In this study, we analyzed the characteristics of soil microbial community structures and their functions in the rhizospheres of white-fleshed dragon fruit (*Hylocereus undatus*) and red-fleshed dragon fruit (*Hylocereus polyrhizus*). The results indicate that the soil bacterial and fungal compositions in the rhizospheres were shaped differently between *H. undatus* and *H. polyrhizus*.

In terms of bacterial community composition, Actinobacteriota, Pseudomonadota, Firmicutes, Chloroflexi, Acidobacteriota, Gemmatimonadota, Bacteroidota, Patescibacteria, and Myxococcota were the common dominant soil bacterial phyla in the rhizospheres of *H. undatus* and *H. polyrhizus*. *Kribbella* and *TM7a* were the unique dominant bacterial genera in the rhizospheres of *H. undatus*, while *Bradyrhizobium* was found as the unique dominant genus in the rhizospheres of *H. polyrhizus*. The LEfSe analysis also showed that *TM7a*, *Novibacillus*, *Cupriavidus,* and *Mesorhizobium* were significantly enriched in the rhizospheres of *H. undatus*, contributing to the significant difference between *H. undatus* and *H. polyrhizus*.

Additionally, at the phylum level, Ascomycota had the highest relative abundance in the rhizospheres of both *H. undatus* and *H. polyrhizus*, but the relative abundance of this phylum in the rhizosphere of *H. polyrhizus* was 1.438 times higher than that in *H. undatus*. As Ascomycota is the most widely studied microbial group in terms of pigment production [52], filamentous ascomycetes fungi have been confirmed as a source of natural pigments. For instance, fungal genera such as *Aspergillus* and *Penicillium* belong to Ascomycota; they can produce natural pigments, ranging from yellow to purple [53,54,55]. Moreover, *Neurospora* is also considered a potential carotenoid producer but does not produce mycotoxins; *Fusarium oxysporum* can produce pink to purple anthranoid pigments [56]. Changes in microbial communities can be induced by plants; they can also provide feedback on plant growth [57]. Particularly, rhizospheric soil microorganisms are closely associated with plant traits [58]. As Ascomycota was abundantly enriched in the rhizospheres of *H. polyrhizus*, it can also be speculated that it is closely related to its betalain-rich characteristic.

Further, at the genus level, *Penicillium* showed the highest relative abundance of any fungal genus in rhizospheres of *H. polyrhizus*, and its relative abundance was 3.048 times higher than that of *H. undatus*. The LEfSe analysis also revealed that *Penicillium*, *Blastobotrys*, *Phialemonium*, *Marasmius,* and *Pseudogymnoascus* were significantly enriched in rhizospheres of *H. polyrhizus*. Previous studies have confirmed that orange-red to red pigments, carotenoids, and betalain could all be derived from *Penicillium* [43,59,60].

Plant varieties not only affect the microbial community structure but also have an impact on microbiome functions [61]. Our results also showed that the expression levels of the top 10 bacterial MetaCyc pathways in the rhizospheres of *H. polyrhizus* were higher than those of *H. undatus*. Meanwhile, higher abundances of saprophytic nutritive fungi, but lower abundances of parasitic fungi, were detected in the rhizospheres of *H. polyrhizus* than those of *H. undatus*. 

## 5. Conclusions

Soil microbial compositions are shaped differently in the rhizospheres of dragon fruits with white (*H. undatus*) and red (*H. polyrhizus*) flesh. Although part of the shared dominant microbial genera could be detected in the rhizospheres of these two dragon fruit genotypes, *Kribbella* and *TM7a* were the unique dominant soil bacterial genera in the rhizospheres of *H. undatus*; in contrast, *Bradyrhizobium* was the unique dominant soil bacterial genus in the rhizospheres of *H. polyrhizus*. Additionally, *Arachniotus* was the unique dominant soil fungal genus in the rhizospheres of *H. undatus*. *Myrothecium* was the unique dominant soil fungal genus in the rhizospheres of *H. polyrhizus*.

## Figures and Tables

**Figure 1 plants-13-01346-f001:**
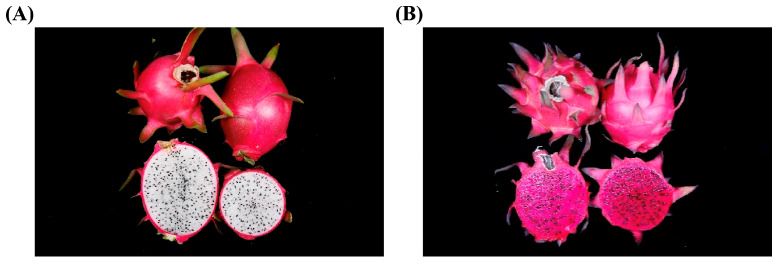
The appearance and morphological characteristics of dragon fruit varieties with white and red flesh. (**A**): Nanning white-flesh (*H. undatus*, RW); (**B**): YuanzhiHong red-flesh (*H. polyrhizus*, RR).

**Figure 2 plants-13-01346-f002:**
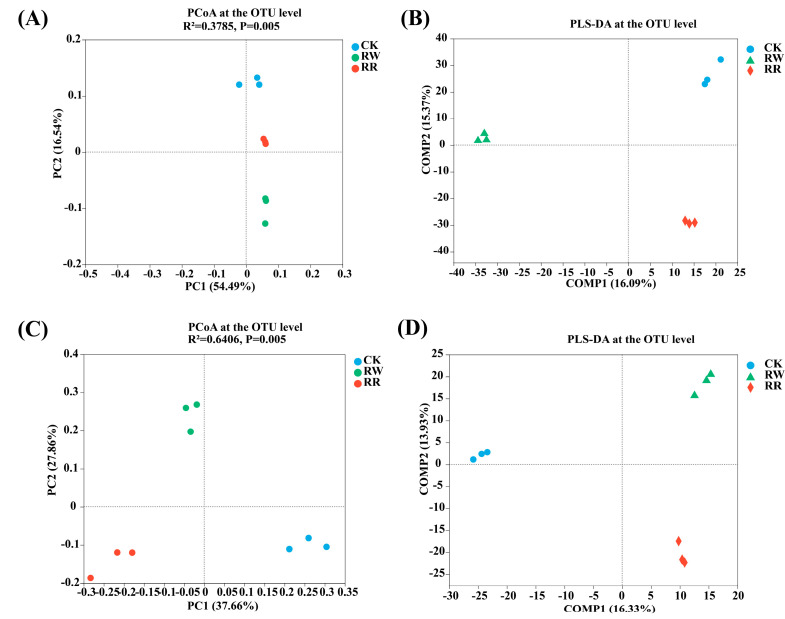
Comparison of the soil bacterial and fungal compositions among RW, RR, and CK. PCoA of soil bacterial (**A**) and fungal (**C**) communities at the OTU level. PLS-DA score plot of the soil bacterial (**B**) and fungal (**D**) communities among RW, RR, and CK.

**Figure 3 plants-13-01346-f003:**
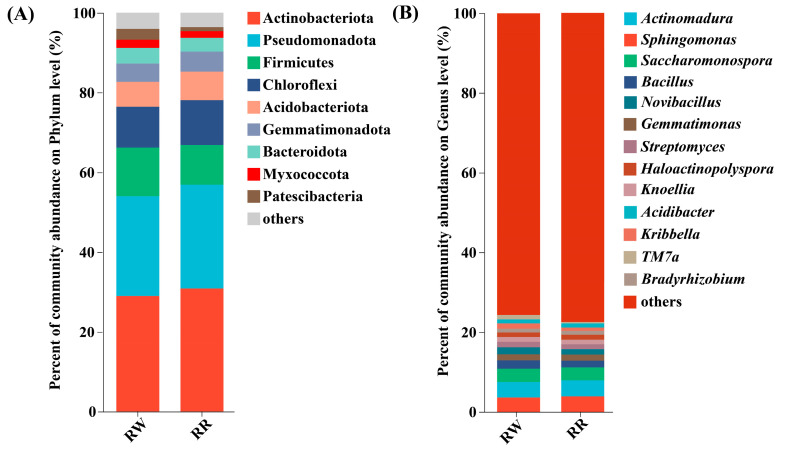
Distributions of dominant soil bacteria in the rhizospheres at the phylum (**A**) and genus (**B**) levels among RW, RR, and CK.

**Figure 4 plants-13-01346-f004:**
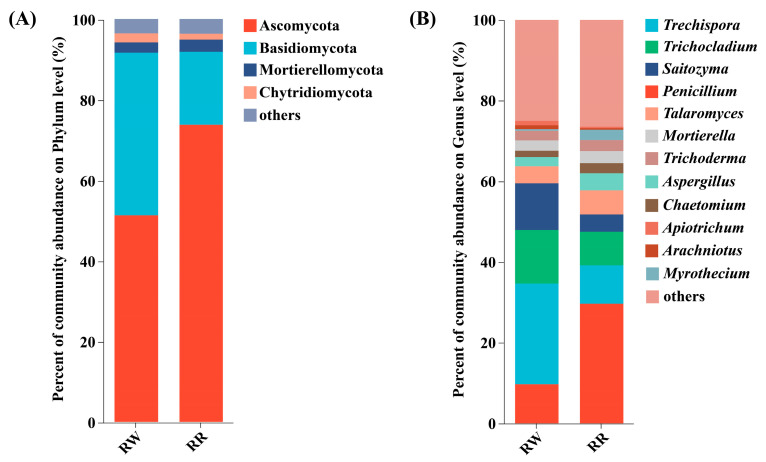
Distributions of dominant soil fungi in the rhizospheres at the phylum (**A**) and genus (**B**) levels among RW, RR, and CK.

**Figure 5 plants-13-01346-f005:**
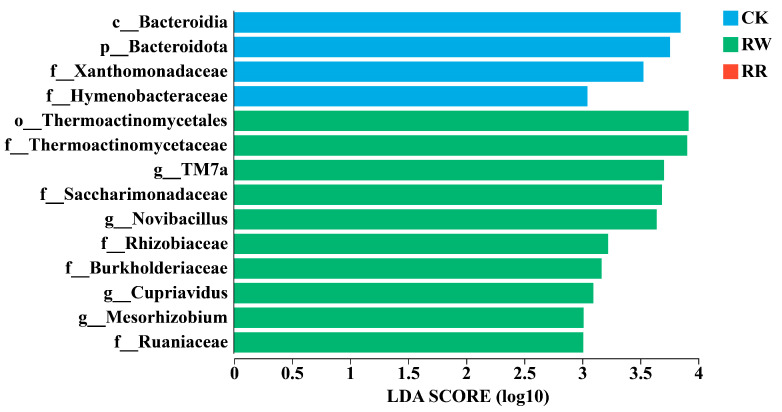
LEfSe analysis of soil bacteria among RW, RR, and CK. Different prefixes indicate different levels (p: phylum; c: class, o: order; f: family; g: genus).

**Figure 6 plants-13-01346-f006:**
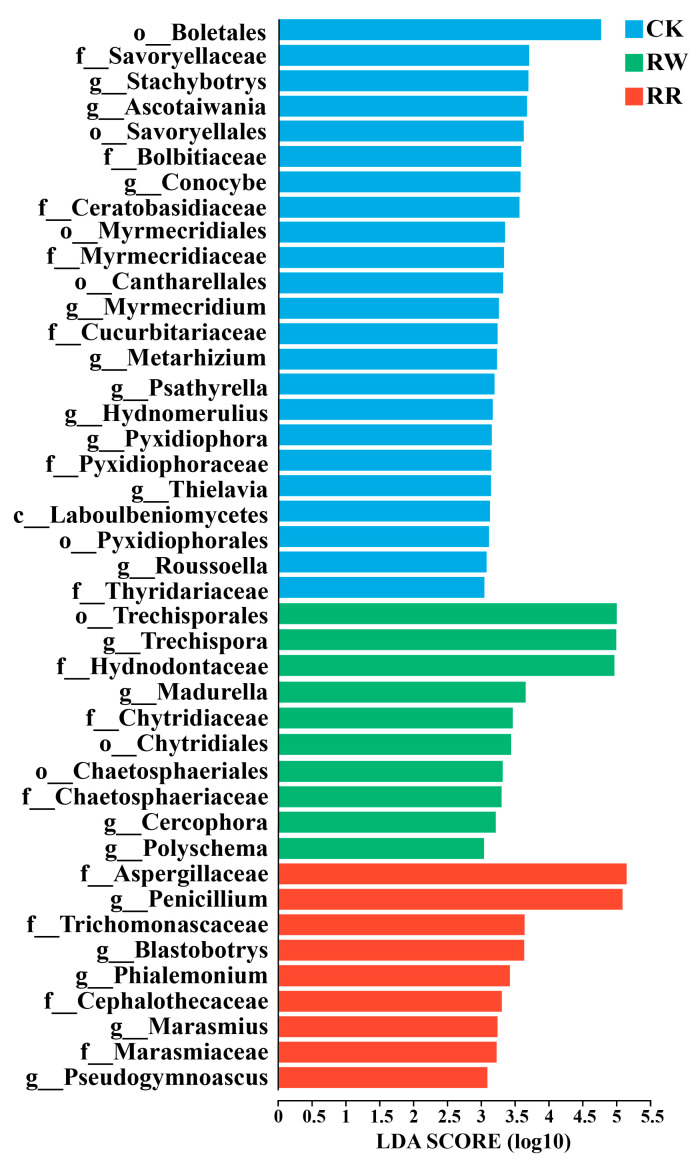
LEfSe analysis of soil fungi among RW, RR, and CK. Different prefixes indicate different levels (p: phylum; c: class, o: order; f: family; g: genus).

**Figure 7 plants-13-01346-f007:**
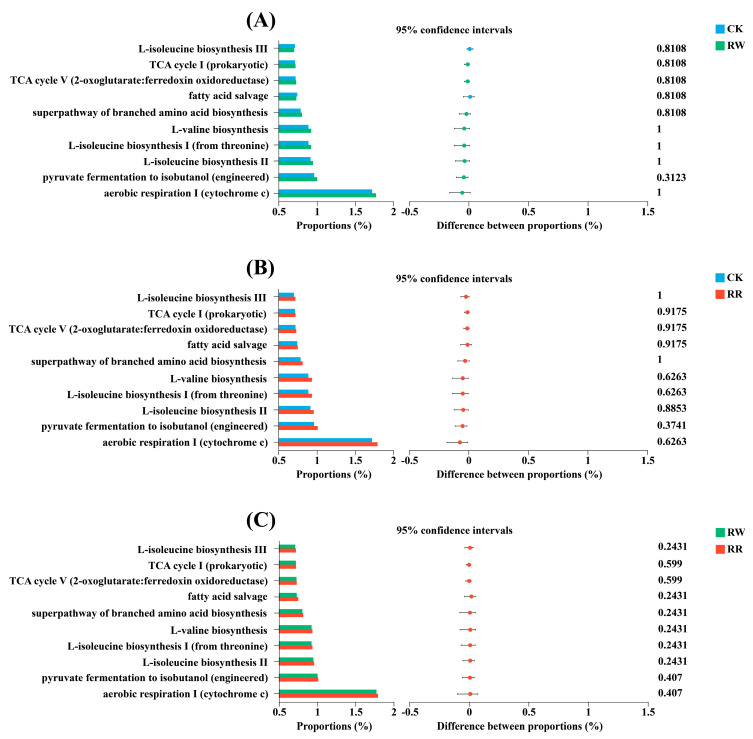
Relative abundance statistics and test for significant differences in soil bacterial expression in the top 10 MetaCycpathway functions among RW, RR, and CK. (**A**): CK vs. RW; (**B**): CK vs. RR; (**C**): RW vs. RR.

**Figure 8 plants-13-01346-f008:**
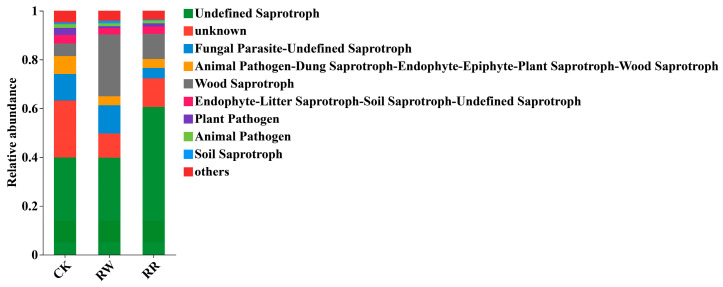
Functional predictions of soil fungal communities among RW, RR, and CK.

**Table 1 plants-13-01346-t001:** Diversity of soil microbes in the rhizospheres of *H. undatus* and *H. polyrhizus*.

Treatment		Shannon	Invsimpson	Ace	Chao	Coverage
Soil bacteria	RW	6.39 ± 0.04 a	183.7 ± 25.89 a	3229 ± 23.96 a	3075 ± 29.39 a	0.98
	RR	6.40 ± 0.03 a	187.4 ± 19.37 a	3120 ± 16.27 a	3189 ± 27.66 a	0.98
	CK	6.29 ± 0.30 a	196.0 ± 60.66 a	3046 ± 200.7 a	3002 ± 169.6 a	0.98
Soil fungi	RW	3.53 ± 0.43 a	10.65 ± 2.391 a	859.3 ± 8.13 a	854.8 ± 26.68 b	0.99
	RR	3.76 ± 0.31 a	8.857 ± 3.796 a	807.8 ± 17.02 a	825.4 ± 13.58 b	0.99
	CK	4.41 ± 0.18 a	26.06 ± 8.993 a	912.6 ± 50.67 a	925.3 ± 37.60 a	0.99

Note: Data in the table are presented as the means ± SDs; values followed by different lowercase letters indicate significant differences at *p* < 0.05 among RW, RR, and CK (RW: *H. undatus*, RR: *H. polyrhizus*, CK: background; the same below).

## Data Availability

The raw data for soil bacterial and fungal sequencing were deposited in the NCBI Sequence Read Archive (SRA) database under accession numbers PRJNA1066732 and PRJNA1066751, respectively.

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
