# Peer review of "How Rhizosphere Microbial Assemblage Is Influenced by Dragon Fruits with White and Red Flesh"

_plants, 2024, doi:10.3390/plants13101346_

Round 1

Reviewer 1 Report

Comments and Suggestions for Authors

Article by Xinyan Zhou et al is dedicated to the determination of the rhizospheric microbial composition of red and yellow dragon fruits.

1) Abstract: Remove all "no rank" taxa and leave only known taxa to make it meaningful for the reader. If the family is classified as "no rank," then take a higher taxon to which it belongs with a standart taxonomic name.

2) Bioinformatics analysis of sequenced reads is not well-described. Which database was used for taxonomy assignment? What similarity threshold was chosen for taxon assignment? At what similarity level were OTUs clustered?

3) Table 1: What does the sample CK mean? What specific background is being referred to?

4) The figures are poorly legible. They need to be enlarged entirely, and the font needs to be increased on some of them.

5) Line 184: Why is it said "in contrast" when the composition is almost identical? The two paragraphs describing the phylum in white and red dragon fruits need to be combined into one and state that at the phylum level, the composition of the microbial community was similar - the main groups were Actinobacteriota (%% in white and red, respectively), Pseudomonadota ...

Please correct Proteobacteria to Pseudomonadota.

6) Line 188: Provide a description of the CK samples in the Materials and Methods section. It's unclear where they suddenly came from.

7) Lines 195-225 need to be rewritten. Provide a paragraph with common genera for all types of samples. State that the difference was not in the names but in their percentage composition. For example, the genus XXX was present in all samples, its abundance varied from %% to %% in such-and-such samples. Then move on to genera that were characteristic only for a particular type of sample. And remove meaningless names like norank_f__JG30-KF-CM45, what biological significance do they have? Replace them with the name of the phylum, and then provide these numbers to indicate who they belong to. The same goes for other "no rank" taxa.

8) Line 266: unclassified_k__Fungi says nothing, it needs to be removed. You can separately add that there was a certain percentage of new organisms in the samples that could only be classified at the kingdom level.

9) Figure 6 is completely illegible.

10) Conclusions: Correct all "no rank" taxa again and include their phylum names.

Some typos:

Line 74: depende

Line 101: rhizospheric should be in lowercase.

Author Response

Response to Reviewer 1 Comments

Article by Xinyan Zhou et al is dedicated to the determination of the rhizospheric microbial composition of red and yellow dragon fruits.

  • Abstract: Remove all "no rank" taxa and leave only known taxa to make it meaningful for the reader. If the family is classified as "no rank," then take a higher taxon to which it belongs with a standart taxonomic name.

Response 1: The authors remove all "norank" and "unclassified" taxa, and rewritten the abstract section according to the Reviewer’s suggestions. We hope that this will make our research more accessible to reviewers and readers alike!

2) Bioinformatics analysis of sequenced reads is not well-described. Which database was used for taxonomy assignment? What similarity threshold was chosen for taxon assignment? At what similarity level were OTUs clustered?

Response 2: Exactly, the authors were apologises for not describing this section clearly. Bacteria and fungi were individually aligned against the Silva (Release 138, http://www.arb-silva.de) database and the Unite (Release 8.0, http://unite.ut.ee/index.php) database for comparative analysis. The non-redundant sequences (excluding single sequences) were clustered into operational taxonomic units (OTUs) based on 97% similarity, and chimeras were removed during the clustering process to obtain representative sequences of the OTUs.A confidence threshold of 0.7 was applied to obtain species classification information corresponding to each OTU. The authors have added this part in manuscript.

3) Table 1: What does the sample CK mean? What specific background is being referred to?.

Response 3: Due to an oversight on the part of the author there is no description of CK. In our study, soil samples from the same management practices without any plant growth were collected as background soil (CK). Because complete sterilisation of a large amount of soil is difficult for us to do, we set up CK with the aim of illustrating the rhizospheric soil microbial community as a result of growing different varieties of dragon fruit.

4) The figures are poorly legible. They need to be enlarged entirely, and the font needs to be increased on some of them.

Response 4: The authors have enlarged all figures entirely. However, due to the formatting requirements of the manuscript, the authors have compressed all the figures slightly, and we are not sure if they are clear enough now. If they are still not clear enough, we will try to find a way to use other formats for the figures.

5) Line 184: Why is it said "in contrast" when the composition is almost identical? The two paragraphs describing the phylum in white and red dragon fruits need to be combined into one and state that at the phylum level, the composition of the microbial community was similar - the main groups were Actinobacteriota (%% in white and red, respectively), Pseudomonadota . Please correct Proteobacteria to Pseudomonadota.

Response 5: The authors have combined the descriptions and rewritten this section. In Figures and manuscript we has corrected Proteobacteria to Pseudomonadota.

6) Line 188: Provide a description of the CK samples in the Materials and Methods section. It's unclear where they suddenly came from.

Response 6: The authors have provided a description of the CK samples in the Materials and Methods section. In conclusion, the rhizospheric soil microbial communities of RW and RR exhibit significant differences compared to background soil (CK), likely due to the recruitment of distinct microbial communities by different dragon fruit varieties. This phenomenon warrants further investigation into the differences in rhizospheric microbial communities between RW and RR. To facilitate a clearer comparison of these microbial communities, the subsequent text will retain information about CK from the figures, but will not provide textual analysis or discussion on the soil microbial community structure of CK.

7) Lines 195-225 need to be rewritten. Provide a paragraph with common genera for all types of samples. State that the difference was not in the names but in their percentage composition. For example, the genus XXX was present in all samples, its abundance varied from %% to %% in such-and-such samples. Then move on to genera that were characteristic only for a particular type of sample. And remove meaningless names like norank_f__JG30-KF-CM45, what biological significance do they have? Replace them with the name of the phylum, and then provide these numbers to indicate who they belong to. The same goes for other "no rank" taxa.

Response 7: The authors remove all "norank" and "unclassified" taxa, and rewritten this part according to the Reviewer’s suggestions. We hope that will make our manuscript more fluent and reasonable.

8) Line 266: unclassified_k__Fungi says nothing, it needs to be removed. You can separately add that there was a certain percentage of new organisms in the samples that could only be classified at the kingdom level.

Response 8: After discussion, the authors decided to delete "unclassified_k__Fungi" and to remove all "unclassified" taxa from the manuscript.

9) Figure 6 is completely illegible.

Response 9: The authors have re-edited all the figures in the manuscript to make them clearer.

10) Conclusions: Correct all "no rank" taxa again and include their phylum names.

Response 10: The authors have rewritten this part according to the Reviewer’s suggestions. 

Some typos: Line 74: depende; Line 101: rhizospheric should be in lowercase..

Response 11: The authors were sorry to make such an simple mistakes. We have corrected the corresponding places in the manuscript.

Reviewer 2 Report

Comments and Suggestions for Authors

The authors determined the microbial community composition of rhizosphere of two dragon fruit varieties in an effort to figure out the cause of the difference in fruit fresh color between them.  Plants affect microbial communities and activities in rhizosphere by excreting carbohydrates and returning dead organic matter to the soil. Both soil microbial community composition and activities show great temporal variation. Thus, the results of one-time sampling could not fully reflect both the dynamics of both parameters. Certain type of bacteria and fungi affect the synthesis of pigment in fruit bodies. However, these are not direct evidence and therefore could not be considered as the direct cause.  Furthermore, a lot of studies showed that fruit color and flavors are mainly controlled by genes. Without assessing the genetic difference between the two varieties, it is not reasonable to ascribe such difference to existence of certain bacterial and fungal genera. 

Comments on the Quality of English Language

English is not good, which makes it hard to understand their results, logics and deduction. 

For example, Lines 11-14, 43, 48 etc.

Author Response

Response to Reviewer 2 Comments

The authors determined the microbial community composition of rhizosphere of two dragon fruit varieties in an effort to figure out the cause of the difference in fruit fresh color between them.  Plants affect microbial communities and activities in rhizosphere by excreting carbohydrates and returning dead organic matter to the soil. Both soil microbial community composition and activities show great temporal variation. Thus, the results of one-time sampling could not fully reflect both the dynamics of both parameters. Certain type of bacteria and fungi affect the synthesis of pigment in fruit bodies. However, these are not direct evidence and therefore could not be considered as the direct cause.  Furthermore, a lot of studies showed that fruit color and flavors are mainly controlled by genes. Without assessing the genetic difference between the two varieties, it is not reasonable to ascribe such difference to existence of certain bacterial and fungal genera.

Response 12: Firstly, thank you very much for your good comments! 

Exactly, we can learn a lot from your valuable comments. However, we still want to discuss with you related some points in this manuscript. As you said, a lot of studies showed that fruit color and flavors are mainly controlled by genes. In fact, we also have the same consideration. There is no doubt that the plant traits are determined by its genotype, our study does not deny this point.

We are inspired by the buildings in human society can be built similarity based on the design drawings, but differences, such as color, shape or size, still can be found for contributing by different workers. Similarity, the plant genes, such as the design drawings, their expressions whether shall be influenced by the “workers (microorganisms) “in environment, too.

Therefore, our aims in this manuscript only want to find out how differences of these workers (microorganisms) possibly related to plant gene expression. i.e., our purpose is identifying that soil microorganisms in rhizospheres whether they are also potentially contributing or relating to the pulp color formation of dragon fruit. Our purposes are not going to deny or disprove that the pulp color are not determined by plant gene expression.

 In our previous studies, we also had found that not only the soil microorganisms in rhizospheres, but also the endophytic microorganisms in crop roots or stems were closely related to crops (watermelon: https://doi.org/10.1038/s41598-022-10533-0 , tomato: https://doi.org/10.1186/s12866-022-02620-z and amaranth: https://doi.org/10.1016/j.sajb.2023.10.035 ) color formations, crops qualities (pumpkins:  https://doi.org/10.3390/microorganisms10081667, https://doi.org/10.3390/microorganisms10112219 and melon: https://doi.org/10.1128/spectrum.04027-22 ) and ripening traits, etc.

English is not good, which makes it hard to understand their results, logics and deduction. For example, Lines 11-14, 43, 48 etc.

Response 13: The authors have asked the professional service for editing the manuscript.

Reviewer 3 Report

Comments and Suggestions for Authors

To date, pitaya is increasingly gaining the public attention due to its high nutritional value and strong antioxidant properties. Especially, betalains pigments synthesized by Hylocereus have high nutritional value and bioactivities. The knowledge on the regulation of their production are still not well understood that's why conditions influencing their biosynthesis are of great imporance. The peer-reviewed mancuscript (plants-2975477) indicates the existence of mycobiomes specific to the described varieties.

General comments:

All drawings must be enlarged and with better quality. In their current form they are illegible.

This chapter "Discussion" is the most controversial because it does not fulfill its functions. It is a summary of the results obtained and not a discussion of them against the background of other studies.

Lack of any introduction to the biosynthesis of the pigment in the plant and the regulation of the genes of this pathway. To do this authors can use doi: 10.3389/fpls.2015.01179. Both varieties described have a full set of genes for the synthesis of betalaine pigments, but the regulation of the expression in each of them affects the final color of the flesh. The authors indicate the involvement of fungal tyrosinases in the production of betalains. The text gives the impression that this is a positive regulation. However, in reality, TYR hydroxylation activity plays equivalent function in H. polyrhizus and H. undatus, and high expression of TYR in H. undatus influences DOPA oxidation and low production of betalaine pigments and as a result of this white flesh. The formation of red flesh is a much further stage involving the involvement of more enzymes, and a key enzyme in color change is cytochrome P450. Therefore, additional fungal tyrosinases should negatively regulate pigment production.

Therefore, the authors should carefully analyze and discuss, based on scientific literature, what mechanisms representatives of fungal species characteristics of the mycobiomes of individual varieties can influence pigment biosynthesis.

Also, Conclusions should not be a repetition of the results, but an indication of what significance these results have for the current state of knowledge or practical applications or how they may contribute to them in the future.

Detailed comments:

l. 14, l.52 when first used, species names should be in full, not abbreviated, versions

l. 34 there is no room for speculation in science

l. 43 ref [3] - This publication is only available in Chinese, please indicate any available in English, e.g.

DOI: 10.5897/AJB09.1603

l. 43. incorrect name of the group of described pigments. Not betaine, but betalain. Please correct in the entire text

l. 44-45 betalain is a pigment and is not a fiber fraction etc. This sentence should be rephrased because it refers to the properties of the pulp.

l. 56 after „after betalain” please give ref.

l. 59 please edit the sentences so that it is clear which microorganisms are bacteria and which are fungi

l. 61 this paragraph is quite chaotic and requires rewording and ordering in terms of style and cause and effect. If we talk about the fact that plant varieties have different microbiomes in one place, etc. Then collect what characteristics the microbiome can give.

l. 96 Was the soil sterilised before the experiment? If so, please specify under what conditions

Comments on the Quality of English Language

Introduction and Discussion chapter might be improved

Author Response

Response to Reviewer 3 Comments

To date, pitaya is increasingly gaining the public attention due to its high nutritional value and strong antioxidant properties. Especially, betalains pigments synthesized by Hylocereus have high nutritional value and bioactivities. The knowledge on the regulation of their production are still not well understood that's why conditions influencing their biosynthesis are of great imporance. The peer-reviewed mancuscript (plants-2975477) indicates the existence of mycobiomes specific to the described varieties.

General comments:

All drawings must be enlarged and with better quality. In their current form they are illegible.

Response 14: The authors have enlarged all figures entirely according to the Reviewer’s suggestions.

This chapter "Discussion" is the most controversial because it does not fulfill its functions. It is a summary of the results obtained and not a discussion of them against the background of other studies.

Response 15: The authors have revised this chapter according to the Reviewer’s suggestions.

Lack of any introduction to the biosynthesis of the pigment in the plant and the regulation of the genes of this pathway. To do this authors can use doi: 10.3389/fpls.2015.01179. Both varieties described have a full set of genes for the synthesis of betalaine pigments, but the regulation of the expression in each of them affects the final color of the flesh. The authors indicate the involvement of fungal tyrosinases in the production of betalains. The text gives the impression that this is a positive regulation. However, in reality, TYR hydroxylation activity plays equivalent function in H. polyrhizus and H. undatus, and high expression of TYR in H. undatus influences DOPA oxidation and low production of betalaine pigments and as a result of this white flesh. The formation of red flesh is a much further stage involving the involvement of more enzymes, and a key enzyme in color change is cytochrome P450. Therefore, additional fungal tyrosinases should negatively regulate pigment production.

Therefore, the authors should carefully analyze and discuss, based on scientific literature, what mechanisms representatives of fungal species characteristics of the mycobiomes of individual varieties can influence pigment biosynthesis.

Response 16: Thank you for your good comments. The authors queried this literature, as well as other relevant literature. It is described in the literature as follows: tyrosine is hydroxylated and oxidised by tyrosinase (TYR) or cytochrome P450 enzyme (CYP76AD) to form cyclic dopa; cyclic dopa is then spontaneously condensed with beet aldolines formed by the action of 4,5-dopa dioxygenase to form betalain. This seems to indicate that tyrosinase has a positive effect on betalain synthesis. The biosynthetic pathway of betalain is very complex and the authors are not yet able to fully understand the pathway of betalain synthesis. So after deliberation, the authors have decided to remove the controversial statement. The authors have rewritten this section of the manuscript in the hope that it will add to the scientific validity of this study, as well as make our manuscript more accessible to reviewers and readers.

Also, Conclusions should not be a repetition of the results, but an indication of what significance these results have for the current state of knowledge or practical applications or how they may contribute to them in the future.

Response 17: The authors have revised this chapter according to the Reviewer’s suggestions.

Detailed comments:

  1. 14, l.52 when first used, species names should be in full, not abbreviated, versions

Response 18: The authors have revised this point according to the Reviewer’s suggestions.

  1. 34 there is no room for speculation in science

Response 19: Thank you for your good comments. The authors have revised this part according to the Reviewer’s suggestions.

  1. 43 ref [3] - This publication is only available in Chinese, please indicate any available in English, e.g. DOI: 10.5897/AJB09.1603

Response 20: The authors have revised this point according to the Reviewer’s suggestions.

  1. 43. incorrect name of the group of described pigments. Not betaine, but betalain. Please correct in the entire text

Response 21: Exactly, the authors were sorry to make such an simple mistakes. We have corrected it in the entire text, thank you very much for the correction.

  1. 44-45 betalain is a pigment and is not a fiber fraction etc. This sentence should be rephrased because it refers to the properties of the pulp.

Response 22: The authors have revised this sentence according to the Reviewer’s suggestions.

  1. 56 after „after betalain” please give ref.

Response 23: The author has consulted the relevant literature and has cited it at the appropriate places in the manuscript.

  1. 59 please edit the sentences so that it is clear which microorganisms are bacteria and which are fungi.

Response 24: The authors have revised this part according to the Reviewer’s suggestions.

  1. 61 this paragraph is quite chaotic and requires rewording and ordering in terms of style and cause and effect. If we talk about the fact that plant varieties have different microbiomes in one place, etc. Then collect what characteristics the microbiome can give.

Response 25: We are very grateful to the reviewers for pointing out the shortcomings. For this reason, we have rewritten this part and asked a professional organisation to edit the manuscript linguistically. 

  1. 96 Was the soil sterilised before the experiment? If so, please specify under what conditions

Response 26: Because complete sterilisation of a large amount of soil is difficult to do, we didn't sterilise the experimental soil. In the meantime, we set up CK (Soil samples from the same management practices without any plant growth) with the aim of illustrating the rhizospheric soil microbial community as a result of growing different varieties of dragon fruits.

Round 2

Reviewer 1 Report

Comments and Suggestions for Authors

Despite the authors making several changes, there are still a few remarks remaining.

Abstract: line 17 and line 22 – repetition of TM7 information

It is incorrect to label bacteria with a relative abundance of 1% as dominant. Anything below 10% should be referred to as minor groups.

Figure 6 is unreadable.

Lines 283-285: rephrase

Line 286-293: try to find potential ecological roles for unique bacterial groups in different dragon fruits.

Author Response

Response to Reviewer 1 Comments

Despite the authors making several changes, there are still a few remarks remaining.

Abstract: line 17 and line 22 – repetition of TM7 information

Response 1: Thank you for your valuable comments, the authors have double checked the abstract section. TM7a in line 17 is the result in the community structure, Meanwhile, in line 22, it is also the result obtained from the LefSe analysis, so it appears twice in the abstract. A note of clarification is in order.

It is incorrect to label bacteria with a relative abundance of 1% as dominant. Anything below 10% should be referred to as minor groups.

Response 2: Thank you for your good comments. The authors also checked and reviewed quite a lot of literatures and found that the relative abundance of microbes at 1% is the common usage in microbial researchs. And then the authors obey this common usage. Meanwhile The authors also searched some relevant literatures and their links attachment fellows for your checks .

https://doi.org/10.1186/s12866-019-1572-x 

https://doi.org/10.1186/s40793-023-00534-5 

https://doi.org/10.3390/plants10122706 

https://doi.org/10.1016/j.catena.2023.107000 

https://doi.org/10.1111/1751-7915.14372 

Figure 6 is unreadable.

Response 3: The authors have re-edited the figure 6 in the manuscript to make it clearer.

Lines 283-285: rephrase

Response 4: The authors have rephrased this part according to the Reviewer’s suggestions.

Line 286-293: try to find potential ecological roles for unique bacterial groups in different dragon fruits.

Response 5: Thank you for your valuable comments. Exactly, try to find potential ecological roles for unique bacterial groups in different dragon fruits shall be good inspirations for future studies.  However, unfortunately, limited to the current technologies, many functions of microbial groups are still unknown, We hope that our findings will be fill the gap in this area!
